# Use of an extended KDIGO definition to diagnose acute kidney injury in patients with COVID-19: A multinational study using the ISARIC–WHO clinical characterisation protocol

Marina Wainstein[1,2]*, Samual MacDonald[3], Daniel Fryer[3], Kyle Young[3], Valeria Balan[4], Husna Begum[5], Aidan Burrell[5,6], Barbara Wanjiru Citarella[7], J. Perren Cobb[8], Sadie Kelly[9], Kalynn Kennon[9], James Lee[10], Laura Merson[11,12], Srinivas Murthy[13], Alistair Nichol[6,14], Malcolm G. Semple[15,16], Samantha Strudwick[9], Steven A. Webb[5], Patrick Rossignol[17], Rolando Claure-Del Granado[18,19], Sally Shrapnel[20,21]*, the ISARIC Clinical Characterisation Group[¶]

1 Faculty of Medicine, University of Queensland, Brisbane, Australia, 2 West Moreton Kidney Health Service, Brisbane, Australia, 3 School of Mathematics and Physics, University of Queensland, Brisbane, Australia, 4 International Severe Acute Respiratory and emerging Infection Consortium, Centre for Tropical Medicine, University of Oxford, Oxford, United Kingdom, 5 Australian and New Zealand Intensive Care Research Centre, Monash University, Melbourne, Australia, 6 The Alfred Hospital, Intensive Care Unit, Melbourne, Australia, 7 ISARIC Global Support Centre, Centre for Tropical Medicine and Global Health, Nuffield Department of Medicine, University of Oxford, Oxford, United Kingdom, 8 University of Southern California, Los Angeles, California, United States of America, 9 Infectious Diseases Data Observatory (IDDO), University of Oxford, Oxford, United Kingdom, 10 Centre for Tropical Medicine and Global Health, Nuffield Department of Medicine, University of Oxford, Oxford, United Kingdom, 11 Infectious Diseases Data Observatory, Centre for Global Health and Tropical Medicine, University of Oxford, Oxford, United Kingdom, 12 International Severe Acute Respiratory and emerging Infections Consortium (ISARIC), Pandemic Sciences Institute, University of Oxford, Oxford, United Kingdom, 13 Faculty of Medicine, University of British Columbia, Vancouver, Canada, 14 University College Dublin Clinical Research Centre at St Vincent's University Hospital, Dublin, Ireland, 15 NIHR Health Protection Research Unit in Emerging and Zoonotic Infections, Institute of Infection Veterinary and Ecological Sciences, University of Liverpool, Liverpool, United Kingdom, 16 Respiratory Unit, Alder Hey Children's Hospital NHS Foundation Trust, Liverpool, United Kingdom, 17 Université de Lorraine, INSERM, Centre d'Investigations Cliniques-Plurithématique 14–33, INSERM U1116, CHRU Nancy, F-CRIN INI-CRCT (Cardiovascular and Renal Clinical Trialists), Nancy, France, 18 Division of Nephrology, Hospital Obrero No 2-CNS, Cochabamba, Bolivia, 19 Universidad Mayor de San Simon, School of Medicine, Cochabamba, Bolivia, 20 Centre for Health Services Research, University of Queensland, Brisbane, Australia, 21 ARC Centre of Excellence for Engineered Quantum Systems, School of Mathematics and Physics, University of Queensland, Brisbane, Australia

¶ Membership of the ISARIC Clinical Characterisation Group is provided in S1 Acknowledgments.
* marinawainstein@outlook.com (MW); s.shrapnel@uq.edu.au (SS)

**Data Availability Statement:** The data that underpin this analysis are highly detailed clinical

## Abstract

### Background

Acute kidney injury (AKI) is one of the most common and significant problems in patients with Coronavirus Disease 2019 (COVID-19). However, little is known about the incidence and impact of AKI occurring in the community or early in the hospital admission. The traditional Kidney Disease Improving Global Outcomes (KDIGO) definition can fail to identify

data on individuals hospitalised with COVID-19. Due to the sensitive nature of these data and the associated privacy concerns, they are available via a governed data access mechanism following review of a data access committee. Data can be requested via the IDDO COVID-19 Data Sharing Platform (www.iddo.org/covid-19). The Data Access Application, Terms of Access and details of the Data Access Committee are available on the website. Briefly, the requirements for access are a request from a qualified researcher working with a legal entity who have a health and/or research remit; a scientifically valid reason for data access which adheres to appropriate ethical principles. The full terms are at https://www.iddo.org/document/covid-19-data-access-guidelines. A small subset of sites who contributed data to this analysis have not agreed to pooled data sharing as above. In the case of requiring access to these data, please contact the ISARIC team at ncov@isaric.org in the first instance who will look to facilitate access.

**Funding:** In the UK this work was supported by grants from: the National Institute for Health Research (NIHR; award CO-CIN-01), the Medical Research Council (MRC; grant MC_PC_19059), the NIHR Health Protection Research Unit in Emerging and Zoonotic Infections at University of Liverpool in partnership with Public Health England (PHE), in collaboration with Liverpool School of Tropical Medicine and the University of Oxford (NIHR award 200907), UK Foreign, Commonwealth and Development Office and Wellcome (215091/Z/18/Z), Bill & Melinda Gates Foundation (OPP1209135). Internationally this work has been supported by the CIHR Coronavirus Rapid Research Funding Opportunity OV2170359, funding by the Health Research Board of Ireland [CTN-2014-12]; the Rapid European COVID-19 Emergency Response research (RECOVER) [H2020 project 101003589] and European Clinical Research Alliance on Infectious Diseases (ECRAID) [965313], the Research Council of Norway grant no 312780, and a philanthropic donation from Vivaldi Invest A/S owned by Jon Stephenson von Tetzchner; Innovative Medicines Initiative Joint Undertaking under Grant Agreement No. 115523 COMBACTE, resources of which are composed of financial contribution from the European Union's Seventh Framework Programme (FP7/2007- 2013) and EFPIA companies, in-kind contribution; is sponsored by INSERM and funded by the REACTing (REsearch & ACtion emergING infectious diseases) consortium and by a grant of the French Ministry of Health (PHRC n°20-0424); Stiftungsfonds zur Förderung der Bekämpfung der Tuberkulose und anderer Lungenkrankheiten of the City of Vienna, Project Number: APCOV22BGM;

patients for whom hospitalisation coincides with recovery of AKI as manifested by a decrease in serum creatinine (sCr). We hypothesised that an extended KDIGO (eKDIGO) definition, adapted from the International Society of Nephrology (ISN) 0by25 studies, would identify more cases of AKI in patients with COVID-19 and that these may correspond to community-acquired AKI (CA-AKI) with similarly poor outcomes as previously reported in this population.

## Methods and findings

All individuals recruited using the International Severe Acute Respiratory and Emerging Infection Consortium (ISARIC)–World Health Organization (WHO) Clinical Characterisation Protocol (CCP) and admitted to 1,609 hospitals in 54 countries with Severe Acute Respiratory Syndrome Coronavirus 2 (SARS-CoV-2) infection from February 15, 2020 to February 1, 2021 were included in the study. Data were collected and analysed for the duration of a patient's admission. Incidence, staging, and timing of AKI were evaluated using a traditional and eKDIGO definition, which incorporated a commensurate decrease in sCr. Patients within eKDIGO diagnosed with AKI by a decrease in sCr were labelled as deKDIGO. Clinical characteristics and outcomes—intensive care unit (ICU) admission, invasive mechanical ventilation, and in-hospital death—were compared for all 3 groups of patients. The relationship between eKDIGO AKI and in-hospital death was assessed using survival curves and logistic regression, adjusting for disease severity and AKI susceptibility. A total of 75,670 patients were included in the final analysis cohort. Median length of admission was 12 days (interquartile range [IQR] 7, 20). There were twice as many patients with AKI identified by eKDIGO than KDIGO (31.7% versus 16.8%). Those in the eKDIGO group had a greater proportion of stage 1 AKI (58% versus 36% in KDIGO patients). Peak AKI occurred early in the admission more frequently among eKDIGO than KDIGO patients. Compared to those without AKI, patients in the eKDIGO group had worse renal function on admission, more in-hospital complications, higher rates of ICU admission (54% versus 23%) invasive ventilation (45% versus 15%), and increased mortality (38% versus 19%). Patients in the eKDIGO group had a higher risk of in-hospital death than those without AKI (adjusted odds ratio: 1.78, 95% confidence interval: 1.71 to 1.80, $p$-value < 0.001). Mortality and rate of ICU admission were lower among deKDIGO than KDIGO patients (25% versus 50% death and 35% versus 70% ICU admission) but significantly higher when compared to patients with no AKI (25% versus 19% death and 35% versus 23% ICU admission) (all $p$-values $<5 \times 10^{-5}$). Limitations include ad hoc sCr sampling, exclusion of patients with less than two sCr measurements, and limited availability of sCr measurements prior to initiation of acute dialysis.

## Conclusions

An extended KDIGO definition of AKI resulted in a significantly higher detection rate in this population. These additional cases of AKI occurred early in the hospital admission and were associated with worse outcomes compared to patients without AKI.

Italian Ministry of Health "Fondi Ricerca corrente–
L1P6" to IRCCS Ospedale Sacro Cuore–Don
Calabria; grants from Instituto de Salud Carlos III,
Ministerio de Ciencia, Spain; Brazil, National
Council for Scientific and Technological
Development Scholarship number 303953/2018-7.
MW declared funding from the University of
Queensland's Research and Training Scholarship.
SM, DF, KY and SS declared funding from Artificial
Intelligence for Pandemics (AI4PAN) at University
of Queensland. SM & SS declared funding from
The Australian Research Council Centre of
Excellence for Engineered Quantum Systems
(EQUS, CE170100009). AN declared funding from
The Health Research Board of Ireland. JL reports
grants from European Commission RECOVER
Grant Agreement No 101003589 and European
Commission ECRAID-Base Grant Agreement
965313. JPC declared funding from US Center for
Disease Control and Prevention Foundation (site
PI, SCCM Discovery-PREP Covid-19 and
influenza), Herrick Medical LLC (industry-
sponsored RCT of iv tubing modification for air-in-
line evacuation, ClinicalTrials.gov NCT04851782.
SK declared funding from Wellcome (222410/Z/21/
Z). MGS reports grants from National Institute of
Health Research UK, Medical Research Council UK,
Health Protection Research Unit in Emerging &
Zoonotic Infections, University of Liverpool. LM
declared funding from the University of Oxford's
COVID-19 Research Response Fund. SM declared
funding from Canadian Institutes of Health
Research. SS declared funding from the University
of Queensland Strategic funding and University of
Queensland Gender Equity Grant. The funders had
no role in study design, data collection and
analysis, decision to publish, or preparation of the
manuscript. All other authors declared no specific
funding for this work.

**Competing interests:** The authors have declared
that no competing interests exist.

**Abbreviations:** ACE2, angiotensin converting
enzyme 2; ACEi, ACE inhibitor; AKI, acute kidney
injury; ARB, angiotensin receptor blocker; CA-AKI,
community-acquired AKI; CCP, Clinical
Characterisation Protocol; CKD, chronic kidney
disease; COVID-19, Coronavirus Disease 2019;
CRF, case report form; eGFR, estimated glomerular
filtration rate; eKDIGO, extended KDIGO; HIC, high
income; ICU, intensive care unit; IQR, interquartile
range; ISARIC, International Severe Acute
Respiratory and Emerging Infection Consortium;
ISN, International Society of Nephrology; KDIGO,
Kidney Disease Improving Global Outcomes;
LLMIC, low and low middle-income countries;
LOS, length of stay; MICE, Multiple Imputation by

## Author summary

### Why was this study done?

- Previous studies have shown that acute kidney injury (AKI) is a common problem among hospitalised patients with Coronavirus Disease 2019 (COVID-19).

- The current biochemical criteria used to diagnose AKI may be insufficient to capture AKI that develops in the community and is recovering by the time a patient presents to hospital.

- The use of an extended definition that can identify AKI both during its development and recovery phase may allow us to identify more patients with AKI. These patients may benefit from early management strategies to improve long-term outcomes.

### What did the researchers do and find?

- In this prospective study, we examined AKI incidence, severity, and outcomes among a large international cohort of patients with COVID-19 using both a traditional and extended definition of AKI.

- We found that the extended definition identified almost twice as many cases of AKI than the traditional definition (31.7% versus 16.8%).

- These additional cases of AKI were generally less severe and occurred earlier in the hospital admission. Nevertheless, they were associated with worse outcomes, including intensive care unit (ICU) admission and in-hospital death (adjusted odds ratio: 1.78, 95% confidence interval: 1.71 to 1.8, $p$-value < 0.001) than those with no AKI.

### What do these findings mean?

- The current definition of AKI fails to identify a large group of patients with AKI that appears to develop in the community or early in the hospital admission.

- Given the finding that these cases of AKI are associated with worse admission outcomes than those without AKI, identifying and managing them in a timely manner are enormously important.

## Introduction

Acute kidney injury (AKI) has been identified as one of most common and significant problems in hospitalised patients with Coronavirus Disease 2019 (COVID-19) [1–3]. Observational studies have consistently shown that patients who develop AKI are more likely to be admitted to an intensive care unit (ICU), require invasive mechanical ventilation, have longer lengths of stay (LOS) and increased mortality [1,2,4]. Autopsy studies point to several potential pathophysiological pathways for AKI including acute tubular injury from hemodynamic shifts, local

Chained Equations; RAS, renin–angiotensin system; RRT, renal replacement therapy; SARS-CoV-2, Severe Acute Respiratory Syndrome Coronavirus 2; sCr, serum creatinine; STROBE, STrengthening the Reporting of OBservational studies in Epidemiology; UMIC, upper middle income; WHO, World Health Organization.

inflammatory and microvascular thrombotic changes from immune dysregulation as well as direct viral invasion through the angiotensin converting enzyme 2 (ACE2) receptor [5].

Until now, most studies looking at AKI in COVID-19 have used the traditional Kidney Disease Improving Global Outcomes (KDIGO) definition, which relies on the rise in serum creatinine (sCr), either by 26.5 μmol/l in 48 hours or by 50% from baseline over a 7-day period [6]. While this definition is likely to adequately capture AKI that develops during a hospital stay, it may fail to identify cases that have developed in the community and are potentially recovering by the time a patient presents to the hospital, thereby underestimating the true incidence of AKI. To address this potential limitation of the KDIGO definition, the International Society of Nephrology (ISN) 0by25 studies added a commensurate fall in sCr to their definition of AKI [7,8]. Using this modified criteria in the feasibility study, it was found that approximately 40% of the community-acquired AKI (CA-AKI) could be identified by a fall in the level of sCr early in the admission, making it a more comprehensive and inclusive definition [8].

The integration of this additional criterion to identify kidney injury has also been highlighted as one of the research priorities in the recent KDIGO report on controversies in AKI [9]. While other papers have indicated the need to revise various aspects of the KDIGO criteria, aside from the 0by25 studies, a decrease in sCr as a marker of AKI has only been explored in infants and neonates, prompting a need for further research in this area [10,11].

Given the global impact of Severe Acute Respiratory Syndrome Coronavirus 2 (SARS-CoV-2) infection across all income and resource settings, combined with the potentially significant burden of AKI occurring in the community, we hypothesised that an extended KDIGO definition, adapted from the ISN's 0by25 studies, would identify more cases of AKI in patients with COVID-19. We also hypothesised that the additional cases identified using this extended criterion may correspond to CA-AKI and be associated with similarly poor outcomes as those shown in previous studies of AKI in COVID-19 [1,2,4].

## Methods

### Study design

The International Severe Acute Respiratory and Emerging Infection Consortium (ISARIC)–World Health Organization (WHO) Clinical Characterisation Protocol (CCP) for Severe Emerging Infections provided a framework for prospective, observational data collection on hospitalised patients affected by pathogens of public health interest. The protocol, case report forms (CRFs), and study information are available online (https://isaric.org/research/covid-19-clinical-research-resources), of which only the core CRF was used in this study [12]. These CRFs were developed to standardise clinical data collection on patients admitted with suspected or confirmed COVID-19 and have been widely used since the start of the pandemic [13,14]. Collection of sCr measurements across all sites was not time standardised, and the frequency of collection was left to the discretion of each site.

This observational study required no change to clinical management and encouraged patient enrollment in other research projects. Protocol and consent forms are available at https://isaric.net/ccp. While written consent was obtained in most cases, for some sites, the local regulators and ethics committees approved oral consent, or waiver of consent, in the context of the pandemic.

The ISARIC–WHO CCP was approved by WHO Ethics Review Committee (RPC571 and RPC572, April 25, 2013). Ethical approval was obtained for each participating country and site according to local requirements (S1 Statement). A prospective analysis plan was used to guide the design of this study [15]. Only the first of the three aims proposed was addressed in this

study, and the addition of an extended AKI definition was adapted from the 0by25 studies in order to better capture AKI occurring in the community or early in the hospital admission [7].

## Study population

**Inclusion and exclusion criteria.** All individuals in the ISARIC–WHO CCP database with clinically diagnosed or laboratory-confirmed SARS-CoV-2 infection admitted to hospital from February 15, 2020 to February 1, 2021 (criteria for clinical diagnosis in S1 Table) were included in this analysis. Patients younger than 18 years of age and those on maintenance renal replacement therapy (RRT; dialysis or transplantation) were excluded. Patients with fewer than two sCr measurements during the admission and those with incomplete or unreliable laboratory data were also excluded (Fig 1).

## AKI definition and diagnosis

AKI was identified biochemically using sCr and incidence rates calculated accordingly. Patients' sCr levels throughout the admission were used to classify them as (i) not having AKI; (ii) having AKI according to the traditional KDIGO definition; or (iii) AKI according to the extended KDIGO (eKDIGO) criteria. For the purpose of this analysis, those patients in the eKDIGO group with AKI diagnosed by a fall in sCr were labelled as deKDIGO (Fig 2). The KDIGO definition of AKI requires a patient to have an increase in sCr by 26.5 μmol/l within 48 hours or an increase to more than 1.5 times the baseline sCr within 7 days [6]. The eKDIGO definition of AKI was adapted from the ISN's 0by25 studies and included a fall in sCr by 26.5 μmol/l within 48 hours or a fall to more than 1.5 times the baseline sCr within 7 days [8]. AKI was then graded according to the corresponding staging criteria for each definition (Table 1). A moving window of 48 hours and 7 days was applied during the entire length of a patient's admission to find the first instance of AKI as well as the highest stage reached. In the case of AKI diagnosis using an increment in sCr, the minimum sCr within that window was deemed as the baseline,

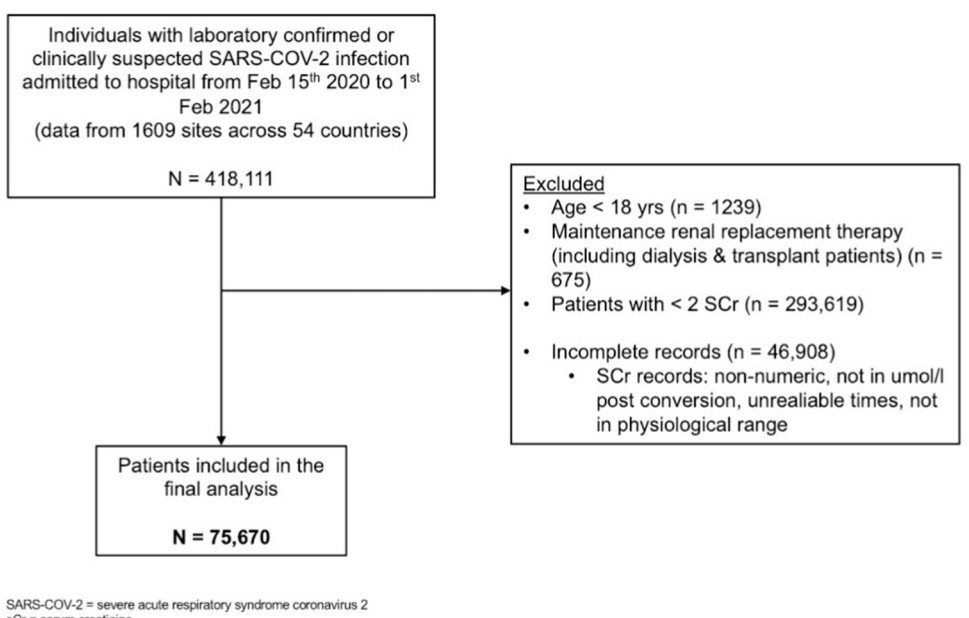

**Fig 1. Flowchart of the study.**

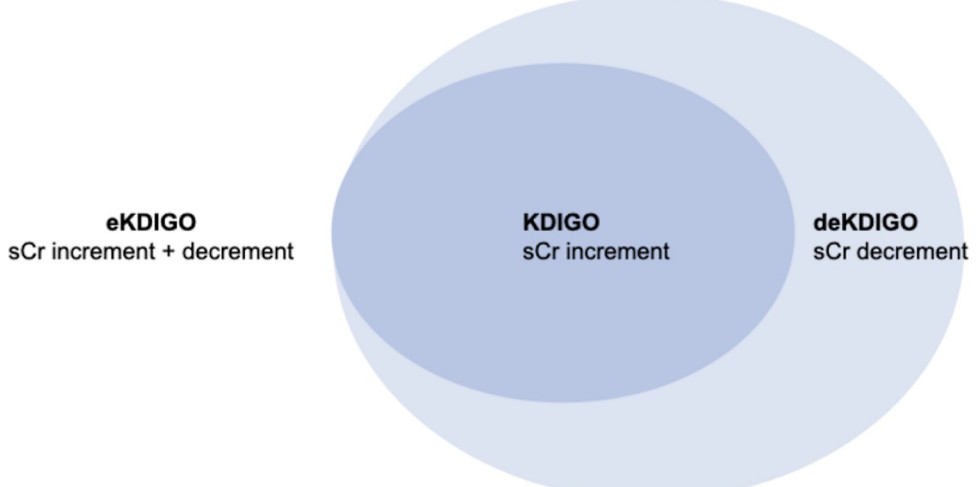

**Fig 2. Visual representation of the relationships between sCr trajectories within the AKI groups (KDIGO, eKDIGO, and deKDIGO).** AKI, acute kidney injury; eKDIGO, extended KDIGO; KDIGO, Kidney Disease Improving Global Outcomes; sCr, serum creatinine.

while for a diagnosis using the decrement, the maximum sCr within that window became the baseline. Information on the timing of acute RRT was not always available so it was not possible to determine whether patients receiving RRT fell into the KDIGO or deKDIGO portions of the eKDIGO group. Given the low likelihood that a patient with a falling creatinine would be given acute RRT, all RRT patients were categorised as being stage 3 AKI within the KDIGO group. Urine volume criteria was not used for either definition as it was not routinely collected in the CRF.

**Table 1. AKI definitions.**

|  | **KDIGO** | **eKDIGO** |
|---|---|---|
| Diagnosis | Increase in sCr by ≥26.5 μmol/l within 48 hours or increase in sCr to ≥1.5 times baseline, which is known or presumed to have occurred within the prior 7 days | Increase in sCr by ≥26.5 μmol/l or decrease in sCr by ≥26.5 μmol/l within 48 hours or increase in sCr to ≥1.5 times baseline or a decrease in sCr to ≥1.5 times baseline, which is known or presumed to have occurred within the prior 7 days* |
| Staging* | | |
| Stage 1 | sCr increase to 1.5 to 1.9 times baseline or increase in sCr by ≥26.5 μmol/l | sCr increase to 1.5 to 1.9 times baseline or an increase in sCr by ≥26.5 μmol/l or sCr decrease to 1.5 to 1.9 times baseline or a decrease by ≥26.5 μmol/l |
| Stage 2 | sCr increase to 2.0 to 2.9 times baseline | sCr increase to 2.0 to 2.9 times baseline or sCr decrease 2.0 to 2.9 times baseline |
| Stage 3 | sCr increase to 3.0 times baseline or sCr increase by ≥353.6 μmol/l or initiation of RRT | sCr increase to 3.0 times baseline or sCr increase by ≥353.6 μmol/l or initiation of RRT or sCr decrease to 3.0 times baseline or sCr decrease to ≥353.6 μmol/l |

*deKDIGO refers to the subgroup of patients diagnosed with AKI by eKDIGO ONLY by the decrease in sCr (see Fig 2).

AKI, acute kidney injury; eKDIGO, extended KDIGO; RRT, renal replacement therapy; sCr, serum creatinine.

### Data collection and time to peak AKI

This study analysed data up until February 15 on patients for whom data collection commenced on or before February 1, 2021. Data were collected and analysed for the duration of a patient's admission. A 14-day rule was applied to focus analysis on individuals who were more likely to have a recorded outcome. By excluding patients enrolled during the last 14 days, we aimed to reduce the number of incomplete data records, thus improving the generalisability of the results and the accuracy of the outcomes.

For both groups (KDIGO and eKDIGO), time to peak AKI from hospital admission and the respective counts for each day were compared by visual inspection of histograms using the first day that a peak stage was reached. From the prespecified data collected in the CRF, information was obtained on demographics and country income level divided according to the World Bank classification (https://data.worldbank.org/country) into high income (HIC), upper middle income (UMIC), and low and low middle-income countries (LLMIC) merged into a single category. Information was obtained on patients' comorbidities and preadmission medications as well as signs and symptoms, observations, and laboratory results on admission. Information collected during the admission included acute treatments, complications, and outcomes. Outcomes included an admission to the ICU, use of invasive mechanical ventilation, and either discharge, transfer to another hospital, in-hospital death, or remaining in hospital. Definitions of all collected variables are provided in S2 Table. A comparison of these variables was performed among patients with eKDIGO AKI and no AKI; deKDIGO and KDIGO AKI; deKDIGO AKI and no AKI.

Patients were classed as lost to follow up if either (a) they were transferred to another facility or (b) they had an unknown outcome and the last date upon which any data were recorded for them was 45 days or before the date of data extraction. Patients with unknown outcome where the last recorded data were less than 45 days old are categorised as receiving ongoing care. Data on readmissions could not be obtained for patients in many countries.

### Statistical analysis

For continuous variables, characteristics were reported as medians and interquartile ranges (IQRs). For categorical variables, counts and percentages were reported. All statistical tests were carried out as pairwise independent samples comparisons. Due to the number of statistical tests conducted, a conservative Bonferroni adjusted significance level of $\alpha_b$ $5 \times 10^{-5}$ was used to limit the study wide probability of a type I error [16]. For continuous variables, the Mann–Whitney U test was used. For categorical variables, Pearson chi-squared test was performed. Missing data were reported as a percentage of the relevant cohort for each variable in Tables 2–4, and further information on its distribution is presented in S3 Table.

A logistic regression was fitted to assess the association between eKDIGO AKI with in-hospital mortality. A *t* test with a significance threshold of 0.001 was used to assess the significance of predictors. Multiple Imputation by Chained Equations (MICE) imputation was used to address variable missingness and a sensitivity analysis showing the results without imputation can be found in S4 Table. Adjustments were made for factors indicating disease severity such as admission to ICU; need for mechanical ventilation; corticosteroid and antifungal treatment; and complicating factors such as bacterial pneumonia, cardiac arrest, coagulation disorders, and rhabdomyolysis. Adjustment was also made for factors known to increase the susceptibility to AKI such as age, sex, diabetes, chronic cardiac disease, chronic pulmonary disease, chronic kidney disease (CKD), hypertension, obesity, and use of renin–angiotensin system (RAS) blockers before admission [6].

**Table 2. Characteristics of patients with no AKI and AKI diagnosed by eKDIGO definition.**

| | | eKDIGO AKI | No AKI | Missingness (%) | *p*-Value |
|---|---|---|---|---|---|
| **Total count** | | | | | |
| | | 23,892 | 51,772 | | |
| **Demographics** | | | | | |
| | Age, year, median (IQR) | 68 (57.5, 78.5) | 67 (53.5, 80.5) | 0 | $p < 5 \times 10^{-5}$ |
| | Female (%) | 8,375 (35) | 22,031 (43) | 0 | $p < 5 \times 10^{-5}$ |
| **Country income level, *n* (%)** | | | | | |
| | HIC | 20,513 (86) | 45,686 (88) | 0 | $p < 5 \times 10^{-5}$ |
| | UMIC | 1,129 (5) | 4,176 (8) | 0 | $p < 5 \times 10^{-5}$ |
| | LLMIC | 2,161 (9) | 1,876 (4) | 0 | $p < 5 \times 10^{-5}$ |
| **AKI grades and RRT, *n* (%)** | | | | | |
| | AKI stage 1 | 13,746 (58) | - | 0 | |
| | AKI stage 2 | 3,682 (15) | - | 0 | |
| | AKI stage 3* | 6,464 (27) | - | 0 | |
| | RRT | 4,252 (19) | - | 9 | |
| **Comorbidities**\*\***, *n* (%)** | | | | | |
| | CKD | 4,059 (18) | 5,433 (11) | 7 | $p < 5 \times 10^{-5}$ |
| | Chronic cardiac disease | 6,225 (27) | 12,772 (25) | 2 | |
| | Chronic pulmonary disease | 2,975 (13) | 6,853 (14) | 7 | |
| | Hypertension | 8,447 (50) | 16,429 (43) | 28 | $p < 5 \times 10^{-5}$ |
| | Dementia | 1,835 (9) | 4,315 (9) | 12 | |
| | Type 2 diabetes | 7,996 (36) | 14,051 (29) | 6 | $p < 5 \times 10^{-5}$ |
| | Liver disease | 842 (4) | 1,653 (3) | 4 | |
| | Malnutrition | 501 (2) | 963 (2) | 12 | |
| | Obesity | 3,792 (19) | 6,404 (15) | 18 | $p < 5 \times 10^{-5}$ |
| **Medications on admission, *n* (%)** | | | | | |
| | NSAIDs | 1,382 (9) | 2,610 (8) | 35 | |
| | ACEis | 2,795 (17) | 5,177 (15) | 33 | $p < 5 \times 10^{-5}$ |
| | ARBs | 1,942 (12) | 3,101 (9) | 33 | $p < 5 \times 10^{-5}$ |
| **Signs and symptoms on admission, *n* (%)** | | | | | |
| | Altered consciousness/confusion | 4,295 (23) | 8,267 (20) | 19 | $p < 5 \times 10^{-5}$ |
| | Diarrhea | 3,740 (20) | 7,983 (19) | 19 | |
| | Fever | 13,333 (64) | 30,320 (66) | 12 | |
| | Vomiting/nausea | 3,387 (18) | 8,099 (19) | 19 | |
| | Muscle aches/joint pain | 3,643 (21) | 8,853 (23) | 25 | $p < 5 \times 10^{-5}$ |
| | Headache | 1,779 (10) | 5,367 (14) | 26 | $p < 5 \times 10^{-5}$ |
| | Sore throat | 1,564 (9) | 3,579 (9) | 27 | |
| | Cough | 12,961 (63) | 29,549 (65) | 12 | |
| | Shortness of breath | 14,824 (72) | 30,334 (66) | 12 | $p < 5 \times 10^{-5}$ |
| | Runny nose | 634 (4) | 1,547 (4) | 28 | |
| **Observations on admission, median (IQR)** | | | | | |
| | Temperature, ˚C | 37.2 (36.2, 38.2) | 37.3 (36.8, 37.8) | 3 | |
| | Systolic BP, mm Hg | 127 (110.5, 143.5) | 130 (115.0, 145.0) | 6 | $p < 5 \times 10^{-5}$ |
| | Diastolic BP, mm Hg | 71 (61.5, 80.5) | 75 (66.0, 84.0) | 7 | $p < 5 \times 10^{-5}$ |
| | Heart rate, BPM | 93 (79.5, 106.5) | 90 (77.5, 102.5) | 7 | $p < 5 \times 10^{-5}$ |
| | Respiratory rate, per minute | 23 (18.5, 27.5) | 21 (17.0, 25.0) | 13 | $p < 5 \times 10^{-5}$ |
| | Oxygen saturation, % | 94 (90.5, 97.5) | 95 (92.5, 97.5) | 7 | $p < 5 \times 10^{-5}$ |
| **Laboratory results on admission, median (IQR)** | | | | | |

(*Continued*)

**Table 2.** (Continued)

|  |  | eKDIGO AKI | No AKI | Missingness (%) | *p*-Value |
|---|---|---|---|---|---|
|  | WBC ($\times 10^9$/L) | 8.2 (5.2, 11.2) | 7 (4.5, 9.5) | 15 | $p < 5 \times 10^{-5}$ |
|  | BUN (mmol/L) | 10.9 (4.9, 16.9) | 6.4 (3.9, 8.9) | 22 | $p < 5 \times 10^{-5}$ |
|  | Potassium (mmol/L) | 4.2 (3.8, 4.6) | 4.1 (3.7, 4.4) | 15 | $p < 5 \times 10^{-5}$ |
|  | CRP (mg/L) | 95.3 (19.8, 170.8) | 69 (12.5, 125.5) | 22 | $p < 5 \times 10^{-5}$ |
|  | sCr (umol/l) | 110 (67.5, 152.5) | 80 (63.0, 97.0) | 11 | $p < 5 \times 10^{-5}$ |
|  | eGFR (ml/min/1.73m$^2$) | 54.4 (29.4, 79.4) | 80.3 (61.8, 98.8) | 12 | $p < 5 \times 10^{-5}$ |
| **Admission treatment, *n* (%)** |  |  |  |  |  |
|  | Antiviral and COVID-19 targeting agents | 5,349 (26) | 9,145 (21) | 16 | $p < 5 \times 10^{-5}$ |
|  | Antibiotic agents | 20,905 (93) | 40,430 (86) | 8 | $p < 5 \times 10^{-5}$ |
|  | Antifungal agents | 2,459 (11) | 2,564 (6) | 11 | $p < 5 \times 10^{-5}$ |
|  | Corticosteroids | 8,553 (38) | 11,905 (25) | 9 | $p < 5 \times 10^{-5}$ |
| **Complications**[**], ***n* (%)** |  |  |  |  |  |
|  | Bacterial pneumonia | 3,827 (19) | 5,761 (13) | 16 | $p < 5 \times 10^{-5}$ |
|  | Cardiac arrest | 1,482 (7) | 988 (2) | 10 | $p < 5 \times 10^{-5}$ |
|  | Coagulation disorder | 1,414 (7) | 1,321 (3) | 15 | $p < 5 \times 10^{-5}$ |
|  | Rhabdomyolysis | 292 (1) | 177 (0.4) | 15 | $p < 5 \times 10^{-5}$ |
| **Outcomes, *n* (%)** |  |  |  |  |  |
|  | ICU admission | 12,579 (54) | 11,652 (23) | 2 | $p < 5 \times 10^{-5}$ |
|  | Invasive mechanical ventilation | 10,264 (45) | 7,294 (15) | 7 | $p < 5 \times 10^{-5}$ |
|  | LOS (median, IQR) | 13 (5.5, 20.5) | 11 (5.0, 17.0) | 4 | $p < 5 \times 10^{-5}$ |
|  | Still in hospital | 1,342 (6) | 1,871 (4) | 2 | $p < 5 \times 10^{-5}$ |
|  | Transferred | 2,052 (9) | 3,457 (7) | 2 | $p < 5 \times 10^{-5}$ |
|  | Discharged | 10,942 (47) | 35,744 (70) | 2 | $p < 5 \times 10^{-5}$ |
|  | Death | 8,890 (38) | 9,794 (19) | 2 | $p < 5 \times 10^{-5}$ |

*Stage 3 includes patients requiring RRT.

**Definitions of comorbidities, complications, and outcomes from the CRFs are presented in S2 Table.

ACEi, angiotensin converting enzyme inhibitor; AKI, acute kidney injury; ARB, angiotensin II receptor blocker; BP, blood pressure; BPM, beats per minute; BUN, blood urea nitrogen; CKD, chronic kidney disease; COVID-19, Coronavirus Disease 2019; CRF, case report form; CRP, C-reactive protein; eGFR, estimated glomerular filtration rate (estimated using the CKD-EPI equation); eKDIGO, extended KDIGO; HIC, high income; ICU, intensive care unit; IQR, interquartile range; LLMIC, low and low middle-income countries; LOS, length of stay; NSAID, nonsteroidal anti-inflammatory drug; RRT, renal replacement therapy; sCr, serum creatinine; UMIC, upper middle income; WBC, white blood cell.

The relationship between eKDIGO AKI and in-hospital death and discharge was described with a survival curve approximated using the Aalen–Johansen estimator, a multistate version of the Kaplan–Meier estimator [17]. The follow-up period began on the day of hospital admission and ended on the day of either discharge or death or 28 days postadmission if no event had occurred. Discharge from hospital was considered an absorbing state (once discharged there was no readmission or death).

All statistical analyses were performed using the R statistical programming language, version 4.0.2 [18,19]. This study is reported as per the STrengthening the Reporting of OBservational studies in Epidemiology (STROBE) guideline (S5 Table).

## Results

From February 15, 2020 to February 1, 2021, data were collected for 418,111 individuals admitted to hospital with clinically suspected or laboratory confirmed SARS-COV-2 infection from 1,609 sites and 54 countries. Of these, 75,670 were used as the analysis cohort (Fig 1).

**Table 3. Characteristics of patients with AKI diagnosed using KDIGO definition versus patients diagnosed with AKI by eKDIGO only by the decrease in sCr (deKDIGO).**

| | | deKDIGO | KDIGO | Missingness (%) | *p*-Value |
|---|---|---|---|---|---|
| **Total count** | | | | | |
| | | 11,188 | 12,704 | | |
| **Demographics** | | | | | |
| | Age, year, median (IQR) | 70 (58.5, 81.5) | 66 (56.5, 75.5) | 0 | $p < 5 \times 10^{-5}$ |
| | Female (%) | 4,259 (38) | 4,116 (33) | 0 | $p < 5 \times 10^{-5}$ |
| **Country income level, *n* (%)** | | | | | |
| | HIC | 10,229 (92) | 10,284 (81) | 0 | $p < 5 \times 10^{-5}$ |
| | UMIC | 333 (3) | 796 (6) | 0 | $p < 5 \times 10^{-5}$ |
| | LLMIC | 600 (5) | 1,561 (12) | 0 | $p < 5 \times 10^{-5}$ |
| **AKI grades and RRT, *n* (%)** | | | | | |
| | AKI stage 1 | 9,169 (82) | 4,577 (36) | 0 | $p < 5 \times 10^{-5}$ |
| | AKI stage 2 | 1,520 (14) | 2,162 (17) | 0 | $p < 5 \times 10^{-5}$ |
| | AKI stage 3 (no RRT) | 499 (4) | 1,713 (13) | 0 | $p < 5 \times 10^{-5}$ |
| | AKI stage 3 (with RRT) | - | 5,965 (47) | 0 | |
| | RRT | - | 4,252 (36) | 9 | |
| **Comorbidities[*], *n* (%)** | | | | | |
| | CKD | 1,797 (17) | 2,262 (19) | 7 | |
| | Chronic cardiac disease | 2,989 (27) | 3,236 (26) | 2 | |
| | Chronic pulmonary disease | 1,599 (15) | 1,376 (12) | 7 | $p < 5 \times 10^{-5}$ |
| | Hypertension | 3,902 (48) | 4,545 (51) | 28 | |
| | Dementia | 1,316 (13) | 519 (5) | 12 | $p < 5 \times 10^{-5}$ |
| | Type 2 diabetes | 3,443 (33) | 4,553 (39) | 6 | $p < 5 \times 10^{-5}$ |
| | Liver disease | 380 (4) | 462 (4) | 4 | |
| | Malnutrition | 284 (3) | 217 (2) | 12 | $p < 5 \times 10^{-5}$ |
| | Obesity | 1,504 (16) | 2,288 (22) | 18 | $p < 5 \times 10^{-5}$ |
| **Medications on admission, *n* (%)** | | | | | |
| | NSAIDs | 627 (8) | 755 (9) | 35 | |
| | ACEis | 1,407 (18) | 1,388 (17) | 33 | |
| | ARBs | 926 (12) | 1,016 (12) | 33 | |
| **Signs and symptoms on admission, *n* (%)** | | | | | |
| | Altered consciousness/confusion | 2,510 (28) | 1,785 (18) | 19 | $p < 5 \times 10^{-5}$ |
| | Diarrhea | 1,885 (21) | 1,855 (18) | 19 | $p < 5 \times 10^{-5}$ |
| | Fever | 6,126 (64) | 7,207 (65) | 12 | |
| | Vomiting/nausea | 1,758 (20) | 1,629 (16) | 19 | $p < 5 \times 10^{-5}$ |
| | Muscle aches/joint pain | 1,617 (20) | 2,026 (22) | 25 | |
| | Headache | 743 (9) | 1,036 (11) | 26 | |
| | Sore throat | 628 (8) | 936 (10) | 27 | $p < 5 \times 10^{-5}$ |
| | Cough | 5,979 (63) | 6,982 (64) | 12 | |
| | Shortness of breath | 6,680 (70) | 8,144 (73) | 12 | $p < 5 \times 10^{-5}$ |
| | Runny nose | 230 (3) | 404 (4) | 28 | $p < 5 \times 10^{-5}$ |
| **Observations on admission, median (IQR)** | | | | | |
| | Temperature, ˚C | 37.3 (36.3, 38.3) | 37.2 (36.2, 38.2) | 3 | |
| | Systolic BP, mm Hg | 123 (107.0, 139.0) | 130 (113.5, 146.5) | 6 | $p < 5 \times 10^{-5}$ |
| | Diastolic BP, mm Hg | 70 (60.5, 79.5) | 72 (62.0, 82.0) | 7 | $p < 5 \times 10^{-5}$ |
| | Heart rate, BPM | 92 (78.5, 105.5) | 93 (79.5, 106.5) | 7 | |
| | Respiratory rate, per min | 22 (17.5, 26.5) | 24 (19.5, 28.5) | 13 | $p < 5 \times 10^{-5}$ |
| | Oxygen saturation, % | 95 (92.0, 98.0) | 94 (91.0, 97.0) | 7 | $p < 5 \times 10^{-5}$ |

*(Continued)*

**Table 3.** (Continued)

| | | deKDIGO | KDIGO | Missingness (%) | *p*-Value |
|---|---|---|---|---|---|
| **Laboratory results on admission, median (IQR)** | | | | | |
| | WBC ($\times 10^9$/L) | 8.1 (5.1, 11.1) | 8.3 (5.3, 11.3) | 15 | |
| | BUN (mmol/L) | 11.6 (5.6, 17.6) | 10.1 (4.1, 16.1) | 22 | $p < 5 \times 10^{-5}$ |
| | Potassium (mmol/L) | 4.1 (3.7, 4.5) | 4.2 (3.8, 4.6) | 15 | |
| | CRP (mg/L) | 89 (20.0, 158.0) | 102.7 (21.2, 184.2) | 22 | $p < 5 \times 10^{-5}$ |
| | sCr, (umol/l) | 119 (80.0, 158.0) | 101 (56.5, 145.5) | 11 | $p < 5 \times 10^{-5}$ |
| | eGFR, ml/min/1.73m$^2$ | 48.6 (28.1, 69.1) | 62.1 (34.6, 89.6) | 12 | $p < 5 \times 10^{-5}$ |
| **Admission treatment, *n* (%)** | | | | | |
| | Antiviral and COVID-19 targeting agents | 2,057 (22) | 3,292 (31) | 16 | $p < 5 \times 10^{-5}$ |
| | Antibiotic agents | 9,718 (91) | 11,187 (94) | 8 | $p < 5 \times 10^{-5}$ |
| | Antifungal agents | 698 (7) | 1,761 (16) | 11 | $p < 5 \times 10^{-5}$ |
| | Corticosteroids | 3,248 (31) | 5,305 (45) | 9 | $p < 5 \times 10^{-5}$ |
| **Complications**[*], ***n*** **(%)** | | | | | |
| | Bacterial pneumonia | 1,461 (15) | 2,366 (22) | 16 | $p < 5 \times 10^{-5}$ |
| | Cardiac arrest | 280 (3) | 1,202 (10) | 10 | $p < 5 \times 10^{-5}$ |
| | Coagulation disorder | 402 (4) | 1,012 (9) | 15 | $p < 5 \times 10^{-5}$ |
| | Rhabdomyolysis | 70 (0.7) | 222 (2) | 15 | $p < 5 \times 10^{-5}$ |
| **Outcomes, *n* (%)** | | | | | |
| | ICU admission | 3,836 (35) | 8,743 (70) | 2 | $p < 5 \times 10^{-5}$ |
| | Invasive mechanical ventilation | 2,596 (24) | 7,668 (63) | 7 | $p < 5 \times 10^{-5}$ |
| | LOS (median, IQR) | 12 (5.5, 18.5) | 15 (6.0, 24.0) | 4 | $p < 5 \times 10^{-5}$ |
| | Still in hospital | 540 (5) | 802 (7) | 2 | $p < 5 \times 10^{-5}$ |
| | Transferred | 924 (8) | 1,128 (9) | 2 | |
| | Discharged | 6,761 (62) | 4,181 (34) | 2 | $p < 5 \times 10^{-5}$ |
| | Death | 2,692 (25) | 6,198 (50) | 2 | $p < 5 \times 10^{-5}$ |

[*]Definitions of comorbidities, complications, and outcomes from the CRFs are presented in S2 Table.

ACEi, angiotensin converting enzyme inhibitor; AKI, acute kidney injury; ARB, angiotensin II receptor blocker; BP, blood pressure; BPM, beats per minute; BUN, blood urea nitrogen; CKD, chronic kidney disease; COVID-19, Coronavirus Disease 2019; CRF, case report form; CRP, C-reactive protein; eGFR, estimated glomerular filtration rate (estimated using the CKD-EPI equation); eKDIGO, extended KDIGO; HIC, high income; ICU, intensive care unit; IQR, interquartile range; KDIGO, Kidney Disease Improving Global Outcomes; LLMIC, low and low middle-income countries; LOS, length of stay; NSAID, nonsteroidal anti-inflammatory drug; RRT, renal replacement therapy; sCr, serum creatinine; UMIC, upper middle income; WBC, white blood cell.

The median length of admission was 12 days (IQR 7, 20 days). Missing data were less than 10% for most variables—although averaging 20% for symptoms on admission—and distributed evenly between groups for those with higher missingness levels (S3 Table).

## Incidence, staging, and timing of peak AKI

With the KDIGO definition 12,704 (16.8%) patients were identified as having AKI during their admission. Using the extended KDIGO definition, a total of 23,892 (31.6%) patients were diagnosed with AKI. A breakdown of the top 10 contributing countries for patients in each group can be found in S1 Fig. The peak stages of AKI with KDIGO and eKDIGO, respectively, were the following: stage 1: 36% and 58%; stage 2: 17% and 15%; and stage 3: 47% and 27%, with a total of 4,252 patients (overall 5.6% of all patients) requiring acute dialysis (Fig 3). Peak sCr occurred more frequently on days 3 and 6 from admission and diminished significantly after day 10 using a KDIGO definition. With the extended definition, an additional 4,019

**Table 4. Characteristics of patients with AKI diagnosed using eKDIGO only by the decrease in sCr (deKDIGO) and no AKI.**

| | | deKDIGO | No AKI | Missingness (%) | *p*-Value |
|---|---|---|---|---|---|
| **Total count** | | | | | |
| | | 11,188 | 51,772 | | |
| **Demographics** | | | | | |
| | Age, year, median (IQR) | 70 (58.5, 81.5) | 67 (53.5, 80.5) | 0 | $p < 5 \times 10^{-5}$ |
| | Female (%) | 4,259 (38) | 22,031 (43) | 0 | $p < 5 \times 10^{-5}$ |
| **Country income level, *n* (%)** | | | | | |
| | HIC | 10,229 (92) | 45,686 (88) | 0 | $p < 5 \times 10^{-5}$ |
| | UMIC | 333 (3) | 4,176 (8) | 0 | $p < 5 \times 10^{-5}$ |
| | LLMIC | 600 (5) | 1,876 (4) | 0 | $p < 5 \times 10^{-5}$ |
| **AKI Grades and RRT, *n* (%)** | | | | | |
| | AKI stage 1 | 9,169 (82) | 0 (0.0) | 0 | |
| | AKI stage 2 | 1,520 (14) | 0 (0.0) | 0 | |
| | AKI stage 3* | 499 (4) | 0 (0.0) | 0 | |
| | RRT | - | - | 9 | |
| **Comorbidities*, *n* (%)** | | | | | |
| | CKD | 1,797 (17) | 5,433 (11) | 7 | $p < 5 \times 10^{-5}$ |
| | Chronic cardiac disease | 2,989 (27) | 12,772 (25) | 2 | |
| | Chronic pulmonary disease | 1,599 (15) | 6,853 (14) | 7 | |
| | Hypertension | 3,902 (48) | 16,429 (43) | 28 | $p < 5 \times 10^{-5}$ |
| | Dementia | 1,316 (13) | 4,315 (9) | 12 | $p < 5 \times 10^{-5}$ |
| | Type 2 diabetes | 3,443 (33) | 14,051 (29) | 6 | $p < 5 \times 10^{-5}$ |
| | Liver disease | 380 (4) | 1,653 (3) | 4 | |
| | Malnutrition | 284 (3) | 963 (2) | 12 | $p < 5 \times 10^{-5}$ |
| | Obesity | 1,504 (16) | 6,404 (15) | 18 | |
| **Medications on admission, *n* (%)** | | | | | |
| | NSAIDs | 627 (8) | 2,610 (8) | 35 | |
| | ACEis | 1,407 (18) | 5,177 (15) | 33 | $p < 5 \times 10^{-5}$ |
| | ARBs | 926 (12) | 3,101 (9) | 33 | $p < 5 \times 10^{-5}$ |
| **Signs and symptoms on admission, *n* (%)** | | | | | |
| | Altered consciousness/confusion | 2,510 (28) | 8,267 (20) | 19 | $p < 5 \times 10^{-5}$ |
| | Diarrhea | 1,885 (21) | 7,983 (19) | 19 | $p < 5 \times 10^{-5}$ |
| | Fever | 6,126 (64) | 30,320 (66) | 12 | |
| | Vomiting/nausea | 1,758 (20) | 8,099 (19) | 19 | |
| | Muscle aches/joint pain | 1,617 (20) | 8,853 (23) | 25 | $p < 5 \times 10^{-5}$ |
| | Headache | 743 (9) | 5,367 (14) | 26 | $p < 5 \times 10^{-5}$ |
| | Sore throat | 628 (8) | 3,579 (9) | 27 | |
| | Cough | 5,979 (63) | 29,549 (65) | 12 | |
| | Runny nose | 230 (3) | 1,547 (4) | 28 | $p < 5 \times 10^{-5}$ |
| **Observations on admission, median (IQR)** | | | | | |
| | Temperature, °C | 37.3 (36.3, 38.3) | 37.3 (36.8, 37.8) | 3 | |
| | Systolic BP, mm Hg | 123 (107.0, 139.0) | 130 (115.0, 145.0) | 6 | $p < 5 \times 10^{-5}$ |
| | Diastolic BP, mm Hg | 70 (60.5, 79.5) | 75 (66.0, 84.0) | 7 | $p < 5 \times 10^{-5}$ |
| | Heart rate, BPM | 92 (78.5, 105.5) | 90 (77.5, 102.5) | 7 | $p < 5 \times 10^{-5}$ |
| | Respiratory rate, per min | 22 (17.5, 26.5) | 21 (17.0, 25.0) | 13 | $p < 5 \times 10^{-5}$ |
| | Oxygen saturation, % | 95 (92.0, 98.0) | 95 (92.5, 97.5) | 7 | $p < 5 \times 10^{-5}$ |
| **Laboratory results on admission, median (IQR)** | | | | | |
| | WBC ($\times 10^{9}$/L) | 8.1 (5.1, 11.1) | 7 (4.5, 9.5) | 15 | $p < 5 \times 10^{-5}$ |
| | BUN (mmol/L) | 11.6 (5.6, 17.6) | 6.4 (3.9, 8.9) | 22 | $p < 5 \times 10^{-5}$ |

(*Continued*)

**Table 4.** (Continued)

|  |  | deKDIGO | No AKI | Missingness (%) | *p*-Value |
|---|---|---|---|---|---|
|  | Potassium (mmol/L) | 4.1 (3.7, 4.5) | 4.1 (3.7, 4.4) | 15 | $p < 5 \times 10^{-5}$ |
|  | CRP (mg/L) | 89 (20.0, 158.0) | 69 (12.5, 125.5) | 22 | $p < 5 \times 10^{-5}$ |
|  | sCr (umol/L) | 119 (80.0, 158.0) | 80 (63.0, 97.0) | 11 | $p < 5 \times 10^{-5}$ |
|  | eGFR, (ml/min/1.73m$^2$) | 48.6 (28.1, 69.1) | 80.3 (61.8, 98.8) | 12 | $p < 5 \times 10^{-5}$ |
| **Admission treatment, *n* (%)** |  |  |  |  |  |
|  | Antiviral and COVID-19 targeting agents | 2,057 (22) | 9,145 (21) | 16 |  |
|  | Antibiotic agents | 9,718 (91) | 40,430 (86) | 8 | $p < 5 \times 10^{-5}$ |
|  | Antifungal agents | 698 (7) | 2,564 (6) | 11 | $p < 5 \times 10^{-5}$ |
|  | Corticosteroids | 3,248 (31) | 11,905 (25) | 9 | $p < 5 \times 10^{-5}$ |
| **Complications**[**], ***n* (%)** |  |  |  |  |  |
|  | Bacterial pneumonia | 1,461 (15) | 5,761 (13) | 16 | $p < 5 \times 10^{-5}$ |
|  | Cardiac arrest | 280 (3) | 988 (2) | 10 |  |
|  | Coagulation disorder | 402 (4) | 1,321 (3) | 15 | $p < 5 \times 10^{-5}$ |
|  | Rhabdomyolysis | 70 (0.7) | 177 (0.4) | 15 |  |
| **Outcomes, *n* (%)** |  |  |  |  |  |
|  | ICU admission | 3,836 (35) | 11,652 (23) | 2 | $p < 5 \times 10^{-5}$ |
|  | Invasive mechanical ventilation | 2,596 (24) | 7,294 (15) | 7 | $p < 5 \times 10^{-5}$ |
|  | LOS (median, IQR) | 12 (5.5, 18.5) | 11 (5.0, 17.0) | 4 | $p < 5 \times 10^{-5}$ |
|  | Still in hospital | 540 (5) | 1,871 (4) | 2 | $p < 5 \times 10^{-5}$ |
|  | Transferred | 924 (8) | 3,457 (7) | 2 | $p < 5 \times 10^{-5}$ |
|  | Discharged | 6,761 (62) | 35,744 (70) | 2 | $p < 5 \times 10^{-5}$ |
|  | Death | 2,692 (25) | 9,794 (19) | 2 | $p < 5 \times 10^{-5}$ |

[*]Does not include RRT.

[**]Definitions of comorbidities, complications, and outcomes from the CRFs are presented in S2 Table.

ACEi, angiotensin converting enzyme inhibitor; AKI, acute kidney injury; ARB, angiotensin II receptor blocker; BP, blood pressure; BPM, beats per minute; BUN, blood urea nitrogen; CKD, chronic kidney disease; COVID-19, Coronavirus Disease 2019; CRF, case report form; CRP, C-reactive protein; eGFR, estimated glomerular filtration rate (estimated using the CKD-EPI equation); HIC, high income; ICU, intensive care unit; IQR, interquartile range; LLMIC, low and low middle-income countries; LOS, length of stay; NSAID, nonsteroidal anti-inflammatory drug; RRT, renal replacement therapy; sCr, serum creatinine; UMIC, upper middle income; WBC, white blood cell.

patients had AKI on day 3 (70% of all AKI diagnosed on that day) and 1,808 on day 6 (64% of all AKI on that day) (Fig 4).

## Demographic and clinical characteristics

Baseline characteristics at hospital admission, acute interventions, complications, and outcomes for patients with AKI diagnosed by eKDIGO versus no AKI, KDIGO versus deKDIGO, and deKDIGO versus no AKI are provided in Tables 2–4, respectively. A majority of patients from all groups were from high-income countries with the highest proportion from LLMICs in the KDIGO group (12% versus 5% in eKDIGO and 9% deKDIGO). Significantly more stage 1 AKI could be seen in the deKDIGO than in KDIGO patients (82% versus 36%), while more severe forms of AKI (stage 3) were dominant in the KDIGO group even after excluding RRT patients (13% versus 4%).

CKD, hypertension, type 2 diabetes, and obesity were significantly more common in patients who developed AKI. The use of ACE inhibitors (ACEis) and angiotensin receptor blockers (ARBs) medications were more common in the KDIGO group as compared to both

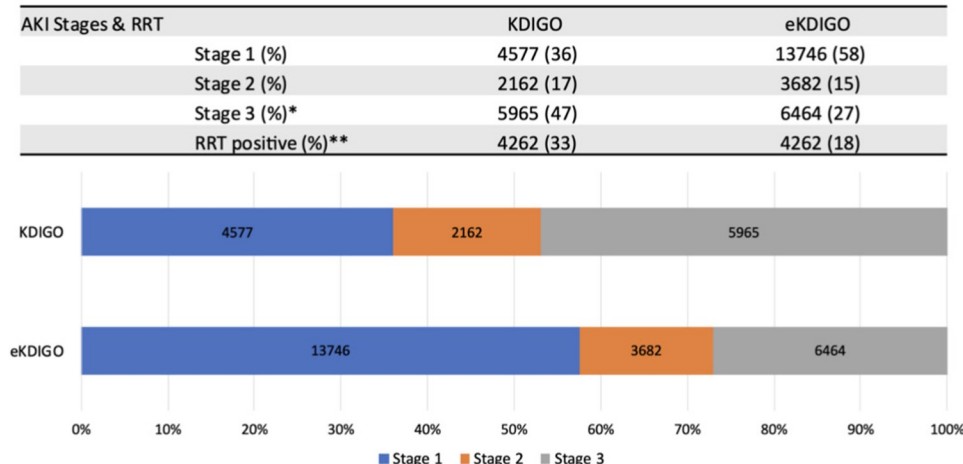

| AKI Stages & RRT | KDIGO | eKDIGO |
|---|---|---|
| Stage 1 (%) | 4577 (36) | 13746 (58) |
| Stage 2 (%) | 2162 (17) | 3682 (15) |
| Stage 3 (%)* | 5965 (47) | 6464 (27) |
| RRT positive (%)** | 4262 (33) | 4262 (18) |

*Includes patients requiring RRT
**Patients requiring RRT during the admission shown as absolute counts and as a proportion of all AKI by each definition.
Abbreviations: RRT = renal replacement therapy; AKI: Acute kidney injury

**Fig 3. Staging of AKI using KDIGO and eKDIGO definitions.** AKI, acute kidney injury; eKDIGO, extended KDIGO; KDIGO, Kidney Disease Improving Global Outcomes; RRT, renal replacement therapy.

the deKDIGO group and the no AKI group. Similarly, administration of antifungal agents and corticosteroids was more common among KDIGO-diagnosed than deKDIGO AKI patients and patients without AKI. Signs, symptoms, and observations at admission were very similar in all groups. At presentation, patients with eKDIGO AKI had a higher blood urea nitrogen (median 10.9 versus 6.4 mmol/l), C-reactive protein (median 95.3 versus 69 mg/L) and sCr (110 umol/l versus 80 umol/l), and lower estimated glomerular filtration rate (eGFR, estimated with CKD-EPI equation) (54 ml/min/1.73m$^2$ versus 80 ml/min/1.73m$^2$) compared to those without AKI. Renal function on admission was worse in the deKDIGO group than in KDIGO

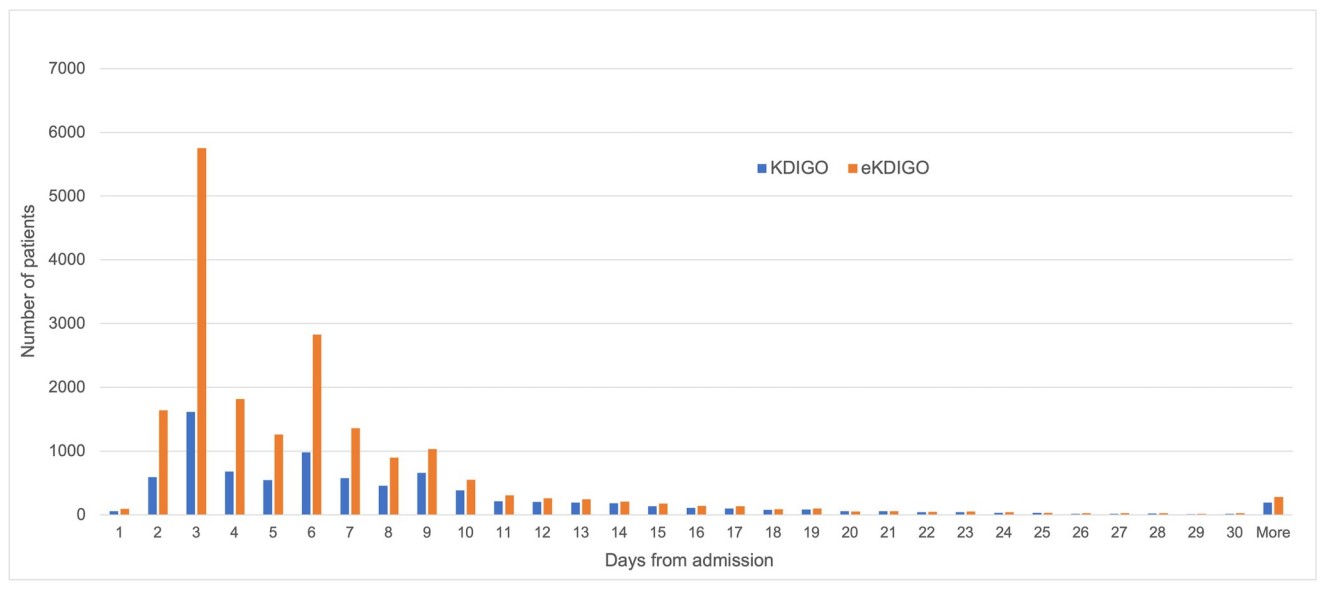

**Fig 4. Day of peak AKI using KDIGO and eKDIGO definitions.** AKI, acute kidney injury; eKDIGO, extended KDIGO; KDIGO, Kidney Disease Improving Global Outcomes.

**Fig 5. Aalen–Johansen survival plot.** Outcomes among patients with AKI diagnosed using eKDIGO criteria and no AKI. Confidence bars are used to illustrate a 95% confidence interval. AKI, acute kidney injury; eKDIGO, extended KDIGO.

patients (eGFR 48 ml/min/1.73m$^2$ versus 62 ml/min/1.73m$^2$). In general, patients with AKI were more likely to have complications during their hospital stay.

## Clinical outcomes

Patients who developed AKI using eKDIGO were more likely to be admitted to the ICU (54%), require invasive mechanical ventilation (45%), and die during their admission (38%) compared to patients without AKI. After adjusting for disease severity, this group of patients had a higher risk of in-hospital death (adjusted odds ratio: 1.77, 95% confidence interval: 1.7–1.85, $p$-value < 0.001) (Table 5), which is further illustrated in the survival curves shown in Fig 5. Patients in the deKDIGO group appeared to have better outcomes and less mortality than those diagnosed by KDIGO criteria, but still had significantly worse outcomes and mortality than patients with no AKI (Tables 3 and 4).

## Discussion

In the largest, multinational cohort of hospitalised patients with COVID-19, it was found that an extended KDIGO criteria for the diagnosis of AKI, which includes a fall in sCr during admission, identified almost twice as many cases of AKI than the traditional KDIGO definition. The majority of these additional cases were stage 1 AKI, occurring early in the admission, supporting the hypothesis that they may represent recovering CA-AKI. This group had comparatively worse outcomes than patients without AKI, making their identification and exploration in future studies enormously important.

The estimated incidence of KDIGO AKI, 16.8%, is consistent with that reported in the first systematic review of AKI in COVID-19 patients [20], while the incidence of eKDIGO AKI fits those studies from the larger New York City cohorts, which had similar rates of ICU admission [2,21]. The mortality rate of 50% among KDIGO-diagnosed AKI patients falls within the range (34% to 50%) reported in previous studies using the same AKI definition [1,2,21–23]. While the inability to exclude readmitted patients may have introduced a degree of survival bias, the fact that readmission rates of less than 3% are seen in other large studies suggests that the effect of this bias is likely to be relatively small [2,21,23].

**Table 5. Logistic regression fitted to assess the association between eKDIGO AKI with in-hospital mortality.**

| Variable | Odds ratio | 95% confidence interval | | p-Value |
|---|---|---|---|---|
| | | Lower | Upper | |
| (Intercept) | 0.051 | 0.048 | 0.054 | <0.001 |
| AKI eKDIGO | 1.776 | 1.705 | 1.85 | <0.001 |
| Female | 0.78 | 0.749 | 0.812 | <0.001 |
| Age 18 to 65 (ref) | 1.0 | - | - | - |
| Age 65 to 85 | 3.969 | 3.778 | 4.17 | <0.001 |
| Age 85+ | 7.773 | 7.285 | 8.294 | <0.001 |
| CKD | 1.317 | 1.248 | 1.39 | <0.001 |
| Chronic cardiac disease | 1.331 | 1.271 | 1.393 | <0.001 |
| Chronic pulmonary disease | 1.514 | 1.436 | 1.595 | <0.001 |
| Hypertension | 0.955 | 0.896 | 1.019 | 0.153 |
| Obesity | 0.904 | 0.841 | 0.972 | 0.008 |
| Type 2 diabetes | 1.153 | 1.096 | 1.214 | <0.001 |
| Preadmission ACEis and ARBs | 0.867 | 0.812 | 0.926 | <0.001 |
| Treatment with corticosteroids | 1.14 | 1.091 | 1.192 | <0.001 |
| Treatment with antifungal agents | 1.218 | 1.138 | 1.305 | <0.001 |
| ICU admission | 1.585 | 1.482 | 1.695 | <0.001 |
| Mechanical ventilation | 2.188 | 2.037 | 2.349 | <0.001 |
| Cardiac arrest | 19.261 | 16.821 | 22.055 | <0.001 |
| Bacterial pneumonia | 1.222 | 1.16 | 1.287 | <0.001 |
| Coagulation disorder | 1.372 | 1.253 | 1.501 | <0.001 |
| Rhabdomyolysis | 1.17 | 0.947 | 1.444 | 0.145 |

MICE imputation used for variable missingness.

ACEi, angiotensin converting enzyme inhibitor; AKI, acute kidney injury; ARB, angiotensin II receptor blocker; CKD, chronic kidney disease; eKDIGO, extended KDIGO; ICU, intensive care unit; MICE, Multiple Imputation by Chained Equations.

In line with what is known regarding AKI susceptibility and sequelae, patients identified in the present study as having AKI—by either definition—were more likely to have CKD, hypertension, and type 2 diabetes mellitus, be on an ACEi or ARB, and generally have more medical complications during their admission than patients who did not develop AKI.

The admission eGFR, sCr, and blood urea nitrogen levels of the eKDIGO AKI population, and specifically those in the deKDIGO group, demonstrated significant impairment early in the admission. This is suggestive of CA-AKI, which would otherwise have gone unrecognised. While these patients had comparatively milder AKI and disease severity than patients in the KDIGO group, they nonetheless incurred significantly more morbidity and mortality than patients without AKI, even after adjusting for confounding factors. With regard to the increased prevalence of stage 1 AKI using the eKDIGO definition, there is growing evidence that even mild episodes of AKI may contribute to the development of CKD [24–26]. This raises the important question of whether this new group of COVID-19 AKI patients would benefit from early management strategies to improve long-term outcomes. Such measures are typically simple—management of fluid balance and removal of nephrotoxic medication for example—and readily implementable, even in resource-poor environments. A follow-up study, similar to the 0by25 feasibility study [8], may be warranted to explore such questions.

The earlier timing of peak AKI in the hospital stay and large proportion of stage 1 cases in the eKDIGO group suggests several possible etiologies. It may point to a prerenal pattern of injury occurring in the setting of dehydration from gastrointestinal fluid losses, fever and

anorexia—a finding supported by the identification of acute tubular injury in autopsy studies of patients with COVID-19 [27]. However, it is also possible that a proportion of these additional, milder, cases of AKI, captured by down trending sCr, are a consequence of early rehydration of patients with either previously normal kidney function or CKD. While the reduced admission eGFR of this group (median 54 ml/min/1.73 m$^2$) makes the former less likely, preadmission sCr measurements would be required to reliably identify the latter. It is reassuring that the proportion of reported CKD in the KDIGO and eKDIGO groups is very similar (19% and 18%, respectively).

It is interesting to consider to what extent the large number of additional cases captured by the extended KDIGO definition are a COVID-specific consequence. While meta-analysis suggests that global estimates of AKI incidence in adult hospitalised patients range between 3% and 18% [28], there are no current estimates of global AKI incidence according to the eKDIGO definition. Moving forward, evaluation of the eKDIGO definition for the diagnosis of AKI in various hospital and community settings will be needed to shed light on whether our findings are particular to a COVID-affected population. In this context, it should be noted that approximately 20% of the analysis cohort had a diagnosis of COVID-19 made on clinical grounds, most likely due to testing shortages and high resource demands during surge phases of the pandemic. While this may have resulted in the potential inclusion of patients with other respiratory illnesses, given that other common respiratory illnesses were notably less prevalent during the pandemic [29,30], it is plausible that a significant proportion of these clinically diagnosed patients did in fact have COVID-19.

This study has some key limitations. The exclusion of patients without 2 sCr measurements may have introduced a degree of selection bias. This could be responsible for the absence of expected geographical differences found between the eKDIGO and no AKI groups and may also have resulted in an underestimation of AKI cases by both definitions [7].

The lack of a time-standardised collection of sCr across all sites also represents a limitation of the study. Patients having more frequent sCr collections may represent a population with more severe illness in whom AKI would be more readily detected, therefore affecting the overall AKI incidence rates and potentially generating a negative survival bias. Nevertheless, it is reassuring that the number of AKI cases are a small proportion of the total sCr collected on any given day (<18%) (S2 Fig), suggesting that the bias introduced by ad hoc sampling was low.

The lack of standardisation in sCr collection may have also affected the reporting of time to peak AKI, the magnitude of peak AKI reached in each individual patient and, in those experiencing both a rise and fall in sCr during their admission, whether AKI was captured during the former phase (KDIGO) or the latter (eKDIGO).

With regard to the distinction between community and hospital acquired AKI, often, a 48-hour threshold is used to identify CA-AKI [31]. Such a definition would preclude many patients in this study who were identified as having AKI on day 3 of admission. It is worth noting that these patients would be identified as CA-AKI (or transient hospital–associated AKI) by the definition proposed by Warnock and colleagues, which integrates sCr trajectories and does not adhere to the somewhat arbitrary 48-hour cutoff [32]. Whether or not the additional cases of AKI captured by eKDIGO are truly reflective of CA-AKI will ultimately require studies that assess this population in a variety of community settings.

To our knowledge, this is the first study to systematically examine an extended KDIGO definition for the identification of AKI against the traditional KDIGO criteria in hospitalised COVID-19 patients. Our population is, as far as we know, the largest and only multinational cohort of patients with COVID-19 from all income country levels. The use of an extended KDIGO definition to diagnose AKI in this population resulted in a significantly higher

incidence rate compared to traditional KDIGO criteria. These additional cases of AKI appear to be occurring in the community or early in the hospital admission and are associated with significantly worse outcomes, highlighting the importance of examining their role and long-term impact in future studies.

## Supporting information

**S1 Statement. Study ethics approval.**
(DOCX)

**S1 Table. Definitions used for clinical COVID-19.** COVID-19, Coronavirus Disease 2019.
(DOCX)

**S2 Table. Definition of comorbidities, complications, and outcomes from the ISARIC CRFs.** ISARIC, International Severe Acute Respiratory and Emerging Infection Consortium; CRF, case report form.
(DOCX)

**S3 Table. Distribution of missingness information between eKDIGO and No AKI patients.** AKI, acute kidney injury; eKDIGO, extended KDIGO.
(DOCX)

**S4 Table. Logistic regression fitted to assess the association between eKDIGO AKI with in-hospital mortality without MICE imputation for variable missingness.** AKI, acute kidney injury; eKDIGO, extended KDIGO; MICE, Multiple Imputation by Chained Equations.
(DOCX)

**S5 Table. STROBE checklist.** STROBE, STrengthening the Reporting of OBservational studies in Epidemiology.
(DOCX)

**S1 Fig. Breakdown of top contributing countries for patients diagnosed with AKI by KDIGO definition (A) and from deKDIGO group (B).** AKI, acute kidney injury; KDIGO, Kidney Disease Improving Global Outcomes.
(DOCX)

**S2 Fig. Number of AKI cases by AKI definition (A = KDIGO and B = eKDIGO) as a proportion of total number of sCrs collected each day.** AKI, acute kidney injury; eKDIGO, extended KDIGO; KDIGO, Kidney Disease Improving Global Outcomes; sCr, serum creatinine.
(DOCX)

**S1 Acknowledgements. The ISARIC Clinical Characterisation Group.** ISARIC, International Severe Acute Respiratory and Emerging Infection Consortium.
(DOCX)

## Acknowledgments

In the UK, this work used data provided by patients and collected by the NHS as part of their care and support #Data Saves Lives. We are extremely grateful to the 2,648 frontline NHS clinical and research staff and volunteer medical students who collected these data in challenging circumstances and the generosity of the patients and their families for their individual contributions in these difficult times. We also acknowledge the support of Jeremy J Farrar and Nahoko Shindo; the coordination in Canada by Sunnybrook Research Institute; endorsement

of the Irish Critical Care- Clinical Trials Group, coordination in Ireland by the Irish Critical Care-Clinical Trials Network at University College Dublin; the COVID clinical management team, AIIMS, Rishikesh, India; Cambridge NIHR Biomedical Research Centre; the dedication and hard work of the Groote Schuur Hospital Covid ICU Team; support by the Groote Schuur nursing and University of Cape Town registrar bodies coordinated by the Division of Critical Care at the University of Cape Town; the dedication and hard work of the Norwegian SARS-CoV-2 study team; Imperial NIHR Biomedical Research Centre; the Firland Foundation, Shoreline, Washington, USA; and the preparedness work conducted by the Short Period Incidence Study of Severe Acute Respiratory Infection.

## Author Contributions

**Conceptualization:** Marina Wainstein, Husna Begum, Aidan Burrell, J. Perren Cobb, Laura Merson, Srinivas Murthy, Alistair Nichol, Malcolm G. Semple, Steven A. Webb, Patrick Rossignol, Rolando Claure-Del Granado, Sally Shrapnel.

**Data curation:** Valeria Balan, Barbara Wanjiru Citarella, Sadie Kelly, Kalynn Kennon, James Lee, Laura Merson, Samantha Strudwick.

**Formal analysis:** Marina Wainstein, Samual MacDonald, Daniel Fryer, Kyle Young, Sally Shrapnel.

**Funding acquisition:** Laura Merson, Sally Shrapnel.

**Investigation:** Marina Wainstein, Samual MacDonald, Daniel Fryer, Kyle Young, Sally Shrapnel.

**Methodology:** Marina Wainstein, Samual MacDonald, Daniel Fryer, Patrick Rossignol, Rolando Claure-Del Granado, Sally Shrapnel.

**Project administration:** Laura Merson.

**Resources:** Laura Merson, Sally Shrapnel.

**Software:** Samual MacDonald, Daniel Fryer, Kyle Young, Sally Shrapnel.

**Supervision:** Sally Shrapnel.

**Validation:** Marina Wainstein, Daniel Fryer, Kyle Young, Sally Shrapnel.

**Visualization:** Marina Wainstein, Samual MacDonald, Daniel Fryer, Kyle Young, Sally Shrapnel.

**Writing – original draft:** Marina Wainstein, Rolando Claure-Del Granado, Sally Shrapnel.

**Writing – review & editing:** Marina Wainstein, Samual MacDonald, Daniel Fryer, Kyle Young, Valeria Balan, Husna Begum, Aidan Burrell, Barbara Wanjiru Citarella, J. Perren Cobb, Sadie Kelly, Kalynn Kennon, James Lee, Laura Merson, Srinivas Murthy, Alistair Nichol, Malcolm G. Semple, Samantha Strudwick, Steven A. Webb, Patrick Rossignol, Rolando Claure-Del Granado, Sally Shrapnel.

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
