## [Editor Report · Decision Letter 0]

26 Nov 2021

Dear Dr Wainstein, 

Thank you for submitting your manuscript entitled "Acute Kidney Injury in Patients with COVID-19 using an Extended KDIGO Definition" for consideration by PLOS Medicine.

Your manuscript has now been evaluated by the PLOS Medicine editorial staff and I am writing to let you know that we would like to send your submission out for external peer review.

Please re-submit your manuscript within two working days, i.e. by Nov 30 2021 11:59PM.

Kind regards,

Callam Davidson

Associate Editor

PLOS Medicine

---

## [Decision Letter · Decision Letter 1]

24 Dec 2021

Dear Dr. Wainstein,

Thank you very much for submitting your manuscript "Acute Kidney Injury in Patients with COVID-19 using an Extended KDIGO Definition" (PMEDICINE-D-21-04861R1) for consideration at PLOS Medicine. 

Your paper was evaluated by a associate editor and discussed among all the editors here. It was also discussed with an academic editor with relevant expertise, and sent to independent reviewers, including a statistical reviewer. The reviews are appended at the bottom of this email and any accompanying reviewer attachments can be seen via the link below:

[LINK]

In light of these reviews, I am afraid that we will not be able to accept the manuscript for publication in the journal in its current form, but we would like to consider a revised version that addresses the reviewers' and editors' comments. Obviously we cannot make any decision about publication until we have seen the revised manuscript and your response, and we plan to seek re-review by one or more of the reviewers. 

We hope to receive your revised manuscript by Jan 25 2022 11:59PM. Please email us (plosmedicine@plos.org) if you have any questions or concerns.

We look forward to receiving your revised manuscript. 

Sincerely,

Callam Davidson, 

Associate Editor

PLOS Medicine

plosmedicine.org

Please revise your title according to PLOS Medicine's style. Your title must be nondeclarative and not a question. It should begin with main concept if possible. "Effect of" should be used only if causality can be inferred, i.e., for an RCT. Please place the study design (e.g. a "a cohort study") in the subtitle (ie, after a colon).

Please structure your abstract using the PLOS Medicine headings (Background, Methods and Findings, Conclusions).

Please combine the abstract Methods and Results sections into one section, “Methods and findings”.

Abstract Background: The final sentence should clearly state the study question.

Abstract Methods and Findings:

* Please include the study design, years during which the study took place, length of follow up, and main outcome measures.

* Please quantify the main results (with 95% CIs and p values).

* Please include the important dependent variables that are adjusted for in the analyses.

Please review the submission guidelines (https://journals.plos.org/plosmedicine/s/submission-guidelines#loc-financial-disclosure-statement) and confirm that your financial disclosure only contains the relevant information.

Please review the submission guidelines (https://journals.plos.org/plosmedicine/s/competing-interests) and confirm that your competing interests statement only contains the relevant information. 

Relevant to the above two comments, it may be worth reviewing previous published articles in PLOS Medicine for an idea of how to present these statements (https://journals.plos.org/plosmedicine/) 

The Data Availability Statement (DAS) requires revision. For the small subset of sites that have not agreed to pooled data sharing, the point of contact for inquiries cannot be a study author.

To facilitate future reviews, please include continuous line numbering throughout your manuscript.

Citations should be in square brackets, and preceding punctuation.

Please define the length of follow up (eg, in mean, SD, and range).

When a p value is given, please specify the statistical test used to determine it.

Please define the abbreviations in Figure 1.

Please provide the unadjusted comparisons as well as the adjusted comparisons in Table 3.

Please specify the variables controlled for in Table 3.

Please provide the name(s) of the institutional review board(s) that provided ethical approval (this can be provided in the supplementary materials if required).

Please specify whether informed consent was written or oral.

Please ensure that the study is reported according to the STROBE guideline, and include the completed STROBE checklist as Supporting Information. 

Please add the following statement, or similar, to the Methods: "This study is reported as per the Strengthening the Reporting of Observational Studies in Epidemiology (STROBE) guideline (S1 Checklist)."

Did your study have a prospective analysis plan? Please state this (either way) early in the Methods section.

Throughout, please ensure Supporting Information files are named and cited per our submission guidelines (https://journals.plos.org/plosmedicine/s/supporting-information).

In the first sentence of the concluding paragraph in your discussion, please temper claims of primacy by stating, "to our knowledge" or something similar. 

Please remove the disclosures and author contributions sections from the end of the main text as this information should be captured in your responses to the submission forms.

References should only use et al after listing the first six authors, and should include dates accessed for internet sources. Please see our website for other reference guidelines https://journals.plos.org/plosmedicine/s/submission-guidelines#loc-references

Comments from the academic editor:

The manuscript addresses a topic of potential interest, which is very timely.

The Authors sough to assess the incidence and impact of acute kidney injury (AKI) occurring in patients with COVID-19 in the community or early after hospital admission using the large ISARIC database and an extended KDIGO definition of AKI. Although the study is very interesting, there are several shortcomings that in the present manuscript preclude sound conclusions, and that should be addressed. To mention some of them:

i) need to provide in the main manuscript the comparison of the outcomes between that part of eKDIGO criteria which otherwise would be missed by standard KDIGO criteria; 

ii) need to provide a table showing differences between AKI diagnosed by deKDIGO and KDIGO criteria, and deKDIGO and no AKI; 

iii) unclear whether there were any readmission, and how the authors treated readmission in the analysis; 

iv) need to explain why in the present manuscript mortality may be lower than that reported in a recent UK study, since the first phase of COVID-19 pandemic was associated with high incidence of AKI and high mortality, especially in Europe; 

v) need to explain in more depth the reasoning behind a decrease in sCr indicating AKI; 

vi) need to clarify the criteria contributing to the number of sCr taken for each individual, and discuss the possible bias of this criteria; 

vii) concern about the fact that the number of sCr measurements/tests taken would appear to affect the probability that a patient would meet the eKDIGO criteria; 

viii) need to clarify and present the typical sCr measurement/test schedules that would also be relevant to the presentation for peak AKI stage; 

ix) need to add for adjustment the age as possible confounder (Table 3); 

x) need to stratify the analysis by age for validation of the findings; 

xi) need to clarify how the confounders (Table 3) were chosen; 

xii) provide a brief sensitivity analysis for the period before vaccination become available; 

xiii) need to stratify analyses by country development; 

xiv) discuss as a study limitation the fact that the involvement of patient without laboratory-confirmed SARS-CoV-2 infection, but with a clinical diagnosis of COVID-19, which relied on symptoms that could be ascribed to other infectious diseases, may have resulted in the inclusion of participants who were hospitalized for non-COVID-19-related conditions; 

xv) need to add in the logistic Generalized Additive Model, several covariates which may have affected the risk of in-hospital mortality, such as ethnicity and co-morbidities (e.g. chronic kidney disease); 

xvi) concern about the conclusions regarding the generalizability of the study findings, since only about 5.3% of the overall patient population was from low- and low/middle-income countries; 

xvii) need to provide the proportion of patients who achieved recovery of kidney function by the time of hospital discharge among surviving patients who developed AKI, according to the traditional and the extended KDIGO definitions.

Comments from the reviewers:

Reviewer #1: This is a very interesting study using eKDIGO criteria for diagnosing AKI 

The clinical features and outcomes of patients diagnosed by falling creatinine in subsequent 7 days is important. Comparison of the outcomes between that part of eKDIGO criteria which otherwise would be missed by standard KDIGO criteria is quite important and that should be in the main manuscript. 

The main point of this study is the use of eKDIGO criteria and the added value of deKDIGO criteria for AKI. Hence, I would recommend table showing differences between AKI diagnosed by deKDIGO and KDIGO criteria and deKDIGO and no AKI in main manuscript. 

Were there any readmissions? We know AKI patients have high rate of readmissions. If so, how did authors treat readmission in the analysis? There is risk of survival bias and how did authors take care of this. Interestingly, the mortality reported by authors in much lower than a UK study and I suspect that this is because of inclusion of readmissions (https://doi.org/10.1371/journal.pmed.1003406). 

Can the authors explain why mortality may be lower as the first phase of COVID pandemic was associated with high incidence of AKI and high mortality, especially in UK and Europe.

I should confess that I am not well versed with GAM and not sure why authors did not perform logistic regression to adjust for the predictors. However, collinearity can still exist in GAM and wonder how did authors check this. Especially, multi-collinearity will exist between mechanical ventilation, cardiac arrest and ICU admission. 

Going forward, it will be important to develop artificial intelligence to electronically detect AKI using the extended criteria in electronic patient record

Reviewer #2: "Acute Kidney Injury in Patients with COVID-19 using an Extended KDIGO Definition" promotes the use of an extended Kidney Disease Improving Global Outcomes (KDIGO) definition, to diagnose acute kidney injury (AKI). The current KDIGO definition depends on whether there is an increase in serum creatinine (sCr) beyond certain thresholds, over 48 hours or within 7 days (Table 1). eKDIGO further includes a decrease in sCr (i.e. deKDIGO) by the same amounts over the same timeframes, as additional criteria. Statistical analysis by pairwise independent samples comparisons adjusting for a number of common confounders suggested that the additionally-identified deKDIGO group would also expect worse outcomes compared to the control group without AKI (sTable5), although not to the extent of the original KDIGO group (sTable4). It is therefore suggested that eKDIGO (which diagnoses the additional deKDIGO group) has the potential to identify more patients with increased mortality (from AKI), that had not been previously diagnosed by the standard KDIGO definition.

This improved determination of risks appears potentially of great value in addressing the ongoing coronavirus pandemic. However, a number of issues might be considered, most importantly those relating to adjustment for confounders:

1. Given the focus on AKI in the study, it might be considered to further clarify the clinical validity of the proposed additions (i.e. decrease in sCr over time) to the KDIGO criteria. While some prior work ([6],[7]) was cited, it might be considered to explain the reasoning behind a decrease in sCr indicating AKI in slightly more depth in the paper, perhaps with empirical metrics/prior research on (expected) sCr variance in health individuals.

2. It is stated that individuals with <2 sCr measurements/tests were excluded, which accounts for n=293,619 of the original n=418,111 individual included in the study. Given that about 70% of the participants has <2 sCr, it might be clarified as to the criteria contributing to the number of sCr taken for each individual. Possible bias of this criteria to the study population might also be discussed further.

3. Moreover, the number of sCr measurements/tests taken would also appear to affect the probability that a patient would meet the (e)KDIGO criteria. For example, if a patient has his sCr increase by 2 times over baseline on the third day after admission, but have sCr decrease to just 1.2 times over baseline by the sixth day, then he would be considered to have AKI had he been tested for sCr around the third day (and possibly other days), but not to have AKI had he only been tested for sCr on the sixth day. sCr testing procedure and descriptive statistics relating to the number of sCr tests (i.e. sampling rate), might thus be discussed/presented.

4. The above clarifications on typical sCr measurement/test schedules would also seem relevant to the presentation of peak AKI stage (Figure 3), since it seems that a patient only has the potential to record a peak AKI on a day when his sCr was taken. Also, it might be clarified as to which day is reflected in Figure 3, if a patient maintaines his peak AKI stage over multiple days (e.g. the first day that the stage is reached)

5. From the confounders used for adjustment (Table 3), age appears a glaring omission, given how significantly it is known to affect mortality rates in particular (to the extent that it might be fair to say that the results may not be meaningful without it). Is there any reason why it was not considered as a confounder, especially as age is included as part of the demographics (Table 2)? Additionally, stratified analysis by age would seem especially relevant for validation of the findings.

6. Related to the above, it might be clarified as to how the confounders (Table 3) were chosen, given the number of available and potentially relevant patient features available from the demographics table (Table 2) alone, that were not included as confounders (e.g. obesity, diabetes, etc.)

7. Since vaccination status would also appear a particularly relevant confounder (for 2021 at least), a brief sensitivity analysis for the period before vaccination became available (if individual vaccination status is unavailable) might be appropriate.

8. Brief stratified analyses by country development (i.e. HIC/UMIC/LLMIC) might be considered.

9. It is stated that missing values were observed for some variables (as described in sTables 3 to 5). The treatment of missing values in the analysis (e.g. through imputation) might be further clarified.

10. There appears potential to further analyze outcomes according to a flexible (e)KDIGO definition, i.e. describe the change in outcomes as the threshold changes from 0.3ml/dl/48 hours, or >=1.5 times/7 days. This might be considered, perhaps in future work.

Reviewer #3: In the present manuscript, the authors evaluated the incidence and impact of AKI occurring in COVID-19 patients in the community or early after hospitalization by means of an extended KDIGO definition of AKI which captures a fall in serum creatinine upon admission. They found that among 75670 COVID-19 patients, the incidence of AKI was about two-fold higher when using the extended KDIGO criteria compared to the traditional KDIGO definition. Most of the additional cases identified were stage I AKI. Patients with AKI defined according to extended KDIGO criteria had worse kidney function at admission, more in-hospital complications and increased mortality risk compared to those without AKI.

The following drawbacks are for the Author's consideration:

1. The present study included patients from the ISARIC database with clinically diagnosed or laboratory-confirmed SARS-CoV-2 infection admitted to hospital from February 15th 2020, to February 1st 2021. Actually, the involvement of patients without laboratory-confirmed SARS-CoV-2 infection, but with a clinical diagnosis of COVID-19, which relied on symptoms (e.g., fever, cough, dyspnoea) that could be ascribed to other infectious diseases, may have resulted in the inclusion of participants who were hospitalized for non-COVID-19-related conditions. This issue should be acknowledged as a study limitation. Moreover, Table 2 lists signs and symptoms of patients at hospital admission, but key symptoms that were used to define clinically diagnosed COVID-19 (Table S1), such as cough, dyspnoea, changes in sense of smell and taste, were not provided. These data should be reported in Table 2.

2. According to the information reported in the Statistical analysis section, comparisons between patients with AKI defined according to the extended KDIGO criteria and those without AKI were performed using a conservative Bonferroni adjusted significance level of 5 x 10-5 (Page 8, lines 19-21). However, in Table 2, which provided characteristics of patients with or without AKI defined according to the extended KDIGO criteria, P values lower than 0.001 were outlined. It should be clarified for which of the comparisons provided in Table 2 P value was lower than 5 x 10-5.

3. The results showed that patients who developed AKI according to the extended KDIGO definition were more likely to require invasive mechanical ventilation and ICU admission compared to those without AKI. In this regard, a recent study found that among patients hospitalized with COVID-19 who required mechanical ventilation and had AKI, 74.5% developed AKI after initiation of mechanical ventilation (Am J Kidney Dis 2021; 77:204-215). Thus, information regarding the temporal relationship of AKI development with initiation of mechanical ventilation and ICU admission should be provided.

4. A logistic Generalized Additive Model was fitted to assess the association between AKI development diagnosed using extended KDIGO criteria with in-hospital mortality, adjusting for gender, requiring ICU admission and/or invasive mechanical ventilation, complications and treatment with corticosteroids or antifungal agents. However, several covariates which may have affected the risk of in-hospital mortality were not included in the model, such as ethnicity, with Black race having been reported to strengthen the association between AKI and the risk of death among patients hospitalized with COVID-19 (Clin J Am Soc Nephrol 2020; 16:14-25), and co-morbidities (e.g., chronic kidney disease). Moreover, the relationship between AKI diagnosed by extended KDIGO criteria and in-hospital mortality should also be described using a Kaplan-Meier survival curve.

5. In the Discussion it was argued that the considerable proportion of the study population from low- and middle-income countries afforded great generalizability of the study findings (Page 16, lines 16-18). Nevertheless, only about 5.3% of the overall patient population was from low- and low middle-income countries (i.e., 4037 out of 75670). Moreover, although 54 countries were involved in the present study, based on data in Figure S1 it can be inferred that more than half of the countries contributed with less than 35 patients with AKI defined according to serum creatinine criteria. Thus, the conclusions regarding the generalizability of the study findings should be tempered.

6. It would be valuable to assess the temporal variation of AKI rate, defined according to the traditional and the extended KDIGO definitions, during the observation period, in light of previous studies which suggested a fall in AKI incidence among COVID-19 patients following the first pandemic wave (Nephrol Dial Transplant 2021; doi: 10.1093/ndt/gfab303; Clin J Am Soc Nephrol 2020; 16:14-25; Kidney Int Rep 2021; 6:916-927).

7. The proportion of patients who achieved recovery of kidney function by the time of hospital discharge among surviving patients who had developed AKI, defined according to the traditional and the extended KDIGO definitions, should be provided.

Minor points: 

- According to the flowchart of the study (Figure 1) individuals with laboratory confirmed SARS-CoV-2 infection admitted to hospital from February 15th 2020 to February 1st 2021 were included in the present study. It should be specified that also patients with clinically diagnosed COVID-19 were included.

- Several inconsistencies throughout the manuscript should be corrected. In particular: i) sixty countries were involved in the study based on the information in the Abstract, but they were 54 according to data in the Results section and in Figure 1; ii) 12740 patients were identified as having AKI according to the traditional KDIGO definition based on the information in the Results section, but they were 12704 according to data in Figure 2 and in Table S4; iii) 23982 patients were identified as having AKI according to the extended KDIGO definition based on the information in the Results section, but they were 23892 according to data in Figure 2 and in Table 2; iv) patients with AKI who required acute dialysis were 4262 according to the information in the Results section, whilst they were 4252 based on data in Table 2; v) the estimated incidence of AKI according to the traditional KDIGO definition was 18% according to the information in the Discussion (Page 14, line 10), but it was 16.8% throughout the manuscript.

- Regarding laboratory parameters, it is unclear whether urea levels were reported as either blood urea concentrations or blood urea nitrogen, and if median values were expressed as mg/dL (as indicated in Table 2) or as mmol/L (as indicated in the main text). As for kidney function, the equation used to estimate glomerular filtration rate (eGFR) should be specified, and it should be clarified whether the reported eGFR values were indexed for body surface area (as reported in the core paper) or not indexed for body surface area (as reported in Table 2).

- The last sentence of the paragraph entitled "Demographic and clinical characteristics" is incomplete.

- Figure S1 was never cited throughout the manuscript; this should be done.

[LINK]

---

## [Decision Letter · Decision Letter 2]

10 Mar 2022

Dear Dr. Wainstein,

Thank you very much for re-submitting your manuscript "Use of an extended KDIGO definition to diagnose acute kidney injury in patients with COVID-19: A prospective, multinational study of the ISARIC cohort." (PMEDICINE-D-21-04861R2) for review by PLOS Medicine.

I have discussed the paper with my colleagues and the academic editor and it was also seen again by three reviewers. I am pleased to say that provided the remaining editorial and production issues are dealt with we are planning to accept the paper for publication in the journal.

[LINK]

We look forward to receiving the revised manuscript by Mar 17 2022 11:59PM.   

Sincerely,

Callam Davidson, 

Associate Editor 

PLOS Medicine

plosmedicine.org

Requests from Editors:

In response to your authorship requests:

1. Any authors above 30 will be included as part of a collaborator group. Please see here for an example - https://pubmed.ncbi.nlm.nih.gov/34890407/. I would suggest that a similar approach would be appropriate here, but please let me know if you have further questions or comments.

2. As you have listed less than 30 authors as part of your preferred upper authorship group, this will not be a problem using the approach outlined above (these authors can be listed first, and any above 30 will be included in the collaborator group). I would suggest using the Acknowledgements section of the manuscript for any further specific acknowledgements. 

For stylistic reasons, please remove the word ‘prospective’ from the title.

Please include the median length of admission (IQR) in the abstract.

Please ensure the main outcome measures are clearly described in the abstract.

Thank you for including a clear and interesting Author Summary. Can I please request the following:

• Please quantify the key findings such as percentages of patients with AKI identified by eKDIGO and KDIGO and the adjusted odds ratio for risk of in-hospital death. 

• Please update the language in the final bullet to refer to associations.

The abbreviations ISN, CI, and RRT appear only once in the abstract, please write these out in full for clarity.

Line 26: Please define ICU on first use in the abstract.

Lines 38-40: You use the term significantly twice here. I believe the first use refers to clinical significance while the second refers to statistical – please clarify to avoid confusion. 

Throughout: Please do not report P<5x10^-5, report instead as <0.001. This applies to tables/figures as well as the main text (Table 5 uses the correct format). 

Line 90: Please define ISN on first use. 

Please ensure Table S5 is labelled as the STROBE (Strengthening the reporting of observational studies in epidemiology) checklist. 

Please confirm the y-axis for the S1 Figure is correct as the intervals are unusual. 

S1 Figure Panel B: The term ‘delta eKDIGO’ appears nowhere else in the manuscript, please confirm this is correct. 

Table 3 is missing a definition for the ** flag in the footnote. 

Line 374: Please update to ‘19% and 18%, respectively’.

The axis labels on S2 Figure are too small to read, please enlarge these. 

References: Journal name abbreviations should be those found in the National Center for Biotechnology Information (NCBI) databases. 

Comments from Reviewers:

Reviewer #1: The authors have responded to all the queries. Though lack of readmission remains a concern, the authors have stated that in limitation

Reviewer #2: We thank the authors for addressing our previous comments, particularly with respect to age as a confounder/target of stratified analysis. While the responses on sCr frequency (e.g. Point 26) are reassuring, it might be considered to perform brief sensitivity analyses on the subset of patients with relatively high sCr frequency, to further confirm the findings.

[LINK]

---

## [Editor Report · Decision Letter 3]

24 Mar 2022

Dear Dr Wainstein, 

On behalf of my colleagues and the Academic Editor, Dr Giuseppe Remuzzi, I am pleased to inform you that we have agreed to publish your manuscript "Use of an extended KDIGO definition to diagnose acute kidney injury in patients with COVID-19: A multinational study using the ISARIC-WHO Clinical Characterisation Protocol" (PMEDICINE-D-21-04861R3) in PLOS Medicine.

PRESS

Sincerely, 

Callam Davidson 

Associate Editor 

PLOS Medicine